# Incorporating the Plant Phenological Trajectory into Mangrove Species Mapping with Dense Time Series Sentinel-2 Imagery and the Google Earth Engine Platform

**Huiying Li [1,2], Mingming Jia [1,3,4,*] , Rong Zhang [1], Yongxing Ren [1,5] and Xin Wen [1,5]**

[1] Key Laboratory of Wetland Ecology and Environment, Northeast Institute of Geography and Agroecology, Chinese Academy of Sciences, Changchun 130102, China; huiying15@mails.jlu.edu.cn (H.L.); zrfighting@163.com (R.Z.); 15271915264@163.com (Y.R.); wenxin18@mails.jlu.edu.cn (X.W.)
[2] School of Management Engineering, Qingdao University of Technology, Qingdao 266520, China
[3] State Key Laboratory of Information Engineering in Surveying, Mapping and Remote Sensing, Wuhan University, Wuhan 430079, China
[4] National Earth System Science Data Center, Beijing 100101, China
[5] College of Earth Sciences, Jilin University, Changchun 130061, China
* Correspondence: jiamingming@iga.ac.cn

**Abstract:** Information on mangrove species composition and distribution is key to studying functions of mangrove ecosystems and securing sustainable mangrove conservation. Even though remote sensing technology is developing rapidly currently, mapping mangrove forests at the species level based on freely accessible images is still a great challenge. This study built a Sentinel-2 normalized difference vegetation index (NDVI) time series (from 2017-01-01 to 2018-12-31) to represent phenological trajectories of mangrove species and then demonstrated the feasibility of phenology-based mangrove species classification using the random forest algorithm in the Google Earth Engine platform. It was found that (i) in Zhangjiang estuary, the phenological trajectories (NDVI time series) of different mangrove species have great differences; (ii) the overall accuracy and Kappa confidence of the classification map is 84% and 0.84, respectively; and (iii) Months in late winter and early spring play critical roles in mangrove species mapping. This is the first study to use phonological signatures in discriminating mangrove species. The methodology presented can be used as a practical guideline for the mapping of mangrove or other vegetation species in other regions. However, future work should pay attention to various phenological trajectories of mangrove species in different locations.

**Keywords:** phenology; species mapping; coastal wetlands; Zhangjiang estuary; *Spartina alterniflora*

## 1. Introduction

Mangrove forests are highly productive ecosystems that maintain coastal ecological balance and biodiversity by providing breeding and nursing grounds for waterfowl, marine, and pelagic species [1–3]. Owing to their intermediate position between the terrestrial and marine environments, mangroves are highly subjected to both natural and anthropogenic disturbances [4]. Thus, for several decades now, mangroves have been extensively studied in studies on sea level rise, ocean surges, carbon storage, and biodiversity conservation [5–7]. While mangrove forests belong to a variety of plant species, different kinds of mangrove species show different ecological functions and adaptive responses to disturbances. Therefore, gathering information on the species composition and distribution of

mangrove forests is essential for the accurate formulation of future studies and management of mangrove ecosystems [8–10].

Remote sensing has served as a sustainable tool in the mapping and monitoring of mangrove forests for decades, primarily because of the logistical and practical difficulties involved in field surveys of the muddy environments [11,12]. Before the launch of the high resolution satellite sensors, it was impossible to accurately discriminate mangrove species with traditional medium resolution satellite data [13]. Recently, with the development of commercial sensors (high spatial resolution, hyperspectral, and active remote sensor), many studies have employed single or combined airborne or satellite imagery to map mangrove species [2,3,8]. However, so far, the mapping of mangrove species with freely accessible imagery still remains a challenge, as mangrove species often exhibit similar spectral signatures and spatial textures [10].

The aforementioned issues indicate the need to explore more substantial features to improve the detection of mangrove species from remote sensing data. A phenological trajectory of plants can be acquired from the time series of remote sensing images through delineating the temporal variation in spectrum during the growing period [14]. To date, remotely sensed plant phenology has been widely used to conduct vegetation discrimination, but most of the studied vegetation has been crops or inland forests [15,16]. Although mangroves are evergreen plants, different mangrove species have different phenophase peaks [17]. However, to date, no studies have dealt with mangrove phenological trajectories in remote sensing-based species mapping [14,18]. So far there has only been one study that has observed mangrove forest phenology using remote sensing data, but the study used 250 m spatial resolution data and focused on the long-term dynamics of mangrove phenological metrics, rather than species mapping [14].

To our knowledge, the lack of phenology-based mangrove species mapping is due to the unavailability of suitable remote sensing imagery. Due to the patchy pattern of mangrove species and the frequent clouds in coastal zones, continuous time series remote sensing data of high quality and fine resolution are difficult to acquire, even commercially. Recently, the Sentinel-2 (S2) MultiSpectral Instrument (MSI) sensor was developed. It has a 10 m spatial resolution and a revisit interval of 2–5 days, which provides the ability to conduct robust and efficient monitoring of the phenological trajectories of different mangrove species. However, the performance of Sentinel imagery in the characterization of mangrove species is unknown. Moreover, how to build and utilize phenological trajectories to discriminate and map mangrove species is still not clear.

In recent years, the cloud-based Google Earth Engine platform (GEE, https://earthengine.google.com) has provided great opportunities to individual geoscientists who are interested in geospatial analysis [16,19]. To our benefit, GEE has introduced unlimited possibilities for phenology-based mangrove species mapping in two aspects: (i) the preprocessed high temporal S2 MSI images can be flexibly accessed; and (ii) a wide range of algorithms made to run geospatial analyses can be remotely operated on Google's supercomputers.

Thus, the aim of this study was to develop a phenology-based strategy to discriminate among and map the geographical distribution of different mangrove species using dense time series S2 MSI imagery and the GEE platform. Specifically, we sought to (i) operate a phenology-based strategy, which could delineate the differences in phenological trajectories between mangrove species and classify different mangrove species and (ii) investigate critical months for separating the seasonal differences between mangrove species.

## 2. Materials and Methods

### 2.1. Study Area

The studied mangrove forest site was the core zone of the Fujian Zhangjiangkou National Mangrove Nature Reserve (FZNNR), which has an area of 2.5 km$^2$ and is located in the estuary of Zhangjiang River, Yunxiao County, Fujian Province, China (Figure 1). In 1992, the reserve was

established by the local government, in 2003, it was approved as one of China's National Nature Reserves, and in 2008, it was classified as a Wetland of International Importance by the Ramsar Convention (Ramsar site no. 1726). The study area is characterized by dominant species of *Kandelia Obovata* (KO), *Aegiceras corniculatum* (AC), and *Avicennia marina* (AM) mixed with a few *Bruguiera gymnorrhiza* and *Acanthus ilicifolius* plants. Recently, the frontier of intertidal zones was rapidly invaded by *Spartina alterniflora* (SA). The climate is a subtropical maritime monsoon climate. The annual average temperature is about 21.2 °C, and the hottest period lies between July and September, while winters are cold with an average temperature of 4 °C. The raining season is from April to September, the average annual precipitation is 1714 mm [20].

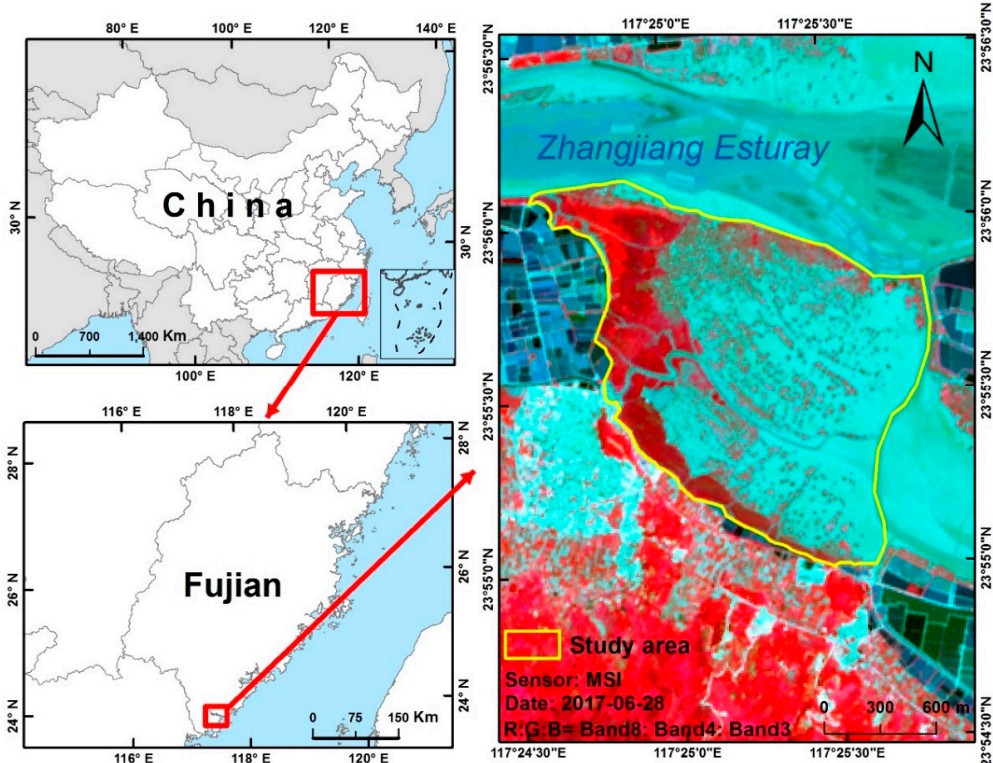

**Figure 1.** The geolocation and overview landscape of the study area in Zhangjiang estuary.

### 2.2. Initial S2 Phenological Dataset

S2, a European Space Agency (ESA) land monitoring mission, has two matching satellites that provide high resolution optical imagery. Sentinel-2A and Sentinel-2B, which operate with the MultiSpectral Instrument (MSI), were successfully launched on June 2015 and March 2017, respectively, and provide important means to augment Earth-observation capabilities [21–24]. These satellites revisit the same place every 2–5 days. The MSI sensor provides 13 spectral bands, with four bands at 10 m, six bands at 20 m, and three bands at 60 m spatial resolution. It is of great utility for a large amount of earth observation applications.

In order to at least cover a whole growing season, two years of MSI images were selected to build the initial phenological dataset. Three steps were operated on the GEE platform as follows:

1.  All the available S2 MSI Level-1C top atmosphere images (S2, radiometric and geometric corrected with sub-pixel accuracy) from 2017 and 2018 were used in this study. These have been archived in the GEE platform as an image collection. In total, there are 199 images in this image collection, which means that each individual pixel represents 199 observations over 24 months. The dense time series observations provide sufficient phenological information for mangrove forests.

2. The QA60 bitmask band, which contains cloud information, was used to mask out opaque and cirrus clouds and scale the S2 quantification value (10,000). Then, a new image collection that excluded clouds or cirrus pixels was returned. We called the new image collection the S2 image collection (S2IC). We called the pixels in S2IC good observations. As we counted, for each individual pixel, the number of good observations ranged from 102 to 111 (Figure 2).

3. The normalized difference vegetation index (NDVI) was calculated for pixels in S2IC. The NDVI, which is the difference between the near-infrared and red bands divided by their sum, is the most commonly used index in studies of global vegetation. Time series changes in NDVI have long been used to represent vegetation phenology [15,25]. In this study, we calculated the NDVI values of each pixel and then built a 10 m spatial resolution time series NDVI image collection, which was called the initial Sentinel phenological dataset (ISPData).

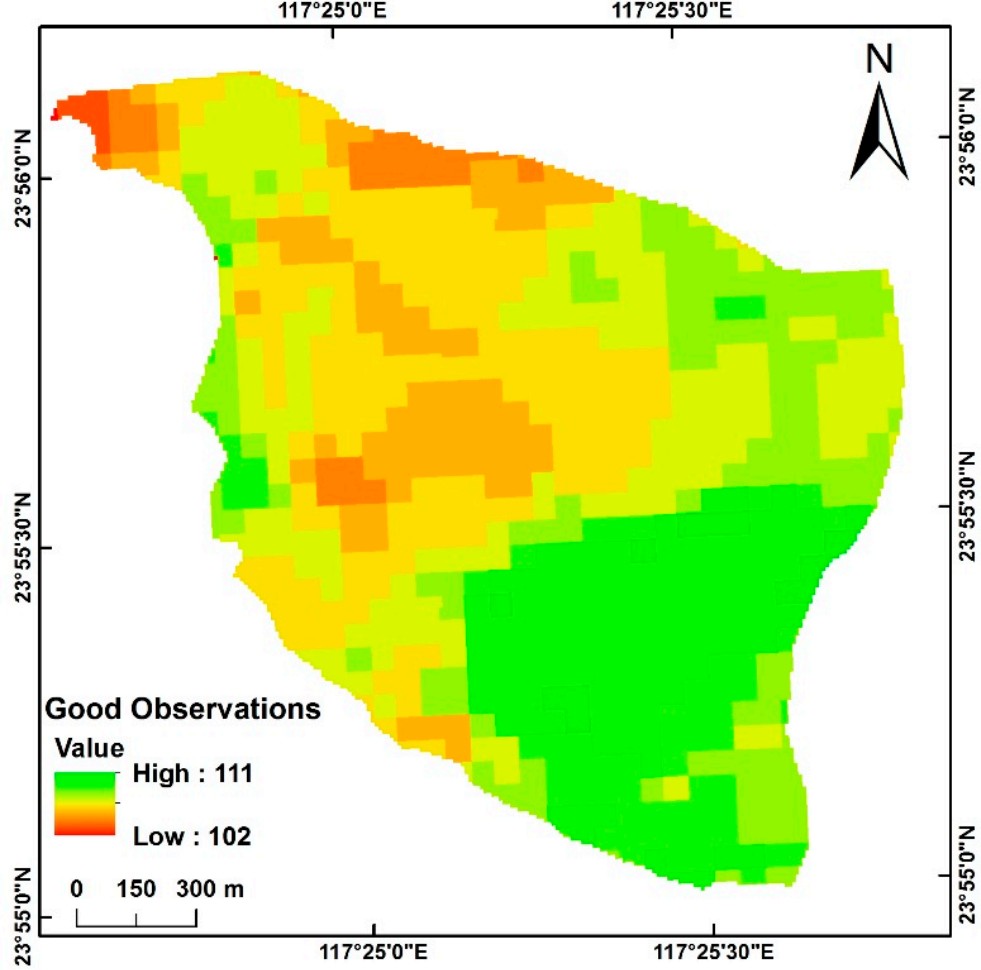

**Figure 2.** Spatial distribution of good observations over the study area from 1 January 2017 to 31 December 2018.

*2.3. Reference Data*

Ground surveys were carried out during December 2018. The location of each sample was measured by the global positioning system (GPS) with an accuracy of less than 1 m, and species names were recorded. With the guidance of a local mangrove export, intertidal mangrove forests were surveyed along walkways as well as regions of mudflats. Due to the muddy environment, we only collected 141 samples during the field survey. Additionally, based on the field surveys and literature research, three main mangrove species were determined to have been discriminated: KO, AM, and AC [26].

In order to get sufficient and accurate training and testing samples, we conducted four unmanned aerial vehicle flights (UAV platform: DJI Phantom 4 pro; flight height ~200 m; flight speed: ~10 m/s) at noon on 29 December 2018, during local low-tide conditions. Then, all of the UAV camera images were an orthomosaic to an image with 5 cm resolution and covered the whole mangrove region (Figure 3A). To select pure training samples with the same spatial resolution as the ISPData, a 10 × 10 m fishnet was built by ArcGIS (Figure 3B). Then, the fishnet was manually adjusted to fit the spatial position of the ISPData pixels (Figure 3(C1,C2)). The adjusted fishnet was adopted to selected pure plots of different land cover types from the UAV orthomosaic image (Figure 3(C3)). At last, based on our field survey and an interview with a local expert, 115, 121, 134, 108, 74, and 105 plots of KO, AM, AC, SA, water (WT), and mudflats (MF) in the adjusted fishnet were selected as training and testing samples, respectively (Figure 4).

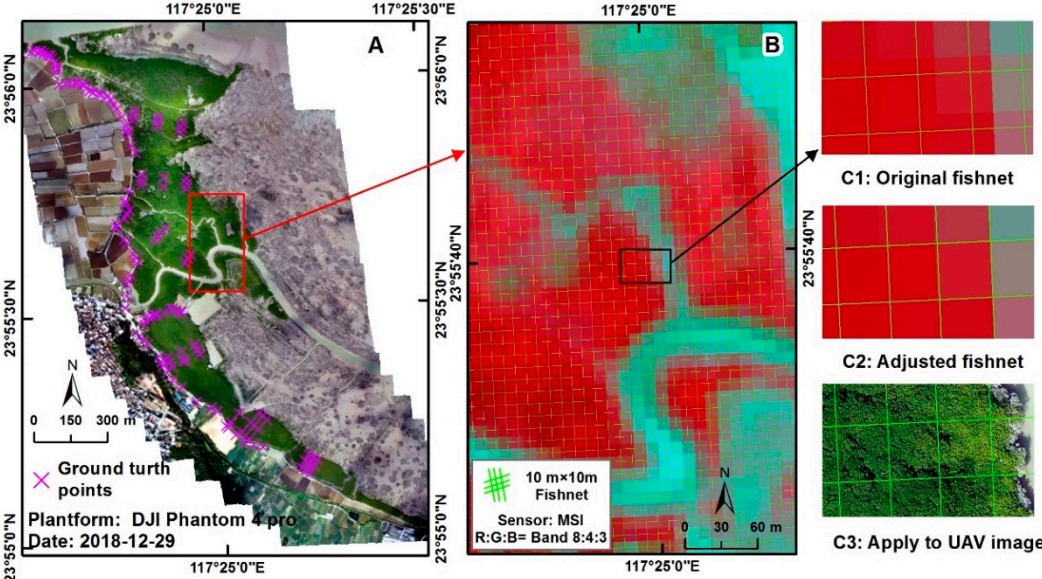

**Figure 3.** Orthomosaic unmanned aerial vehicle (UAV) image (**A**) and fishnet used to select samples (**B**). C1: original fishnet; C2: adjusted fishnet which fit pixels into the MSI image; and C3: application of the adjusted fishnet to the UAV images.

### 2.4. Methods

There were three steps to obtain our aim. First, a high-quality time series dataset was built to support species detection strategies. Second, the random forest classification algorithm was employed to classify mangrove species. Third, crucial months for characterizing the phenology variability of mangrove species were explored.

### 2.4.1. Building High-Quality Sentinel Phenological Dataset (HSPData)

There were nearly always disturbances in time series of NDVI. Although we masked the clouds in ISPData, there was still atmospheric variability and other effects. These uncertainties greatly affected the trajectory of phenology, thus showed up as undesirable noise. Moreover, after masking the clouds, there were many data gaps in the ISPData. To build a high quality phenological dataset, the harmonic analysis of time series (HANTS) algorithm was adopted in this study [27]. HANTS accomplishes two tasks: firstly, the removal and smoothing of noises in time series observations; secondly, interpolation certain values to fill the gaps of inconsecutive time series observations. The theory of HANTS is to build a time series model based on Fourier series, meanwhile, discriminating outliers involved in the time series model. This method excludes the outliers and replaces them by the values simulated by the Fourier series. After applying HANTS to ISPData in GEE platform, a high-quality Sentinel

phenological dataset (HSPData) was constructed. Figure 5 shows two NDVI time series curves of a randomly selected pixel before (ISPData) and after (HSPData) HANTS application, respectively. Figure 6 shows the typical NDVI time series profiles of KO, AC, AM, and SA. The volatilities in time series NDVI profile of mangroves could be mainly explained by litterfall dynamics. The three main mangrove species had different phenological trajectories. KO shows obvious seasonal trends, because it had a high shedding rate. AM and AC had a similar shedding rate, but the NDVI values of AC were always higher than the AM's. The difference of NDVI time series in various species was significant due to the capacity of cold resistance and tolerance of lack water during the grow seasons, which was beneficial to identify mangroves species.

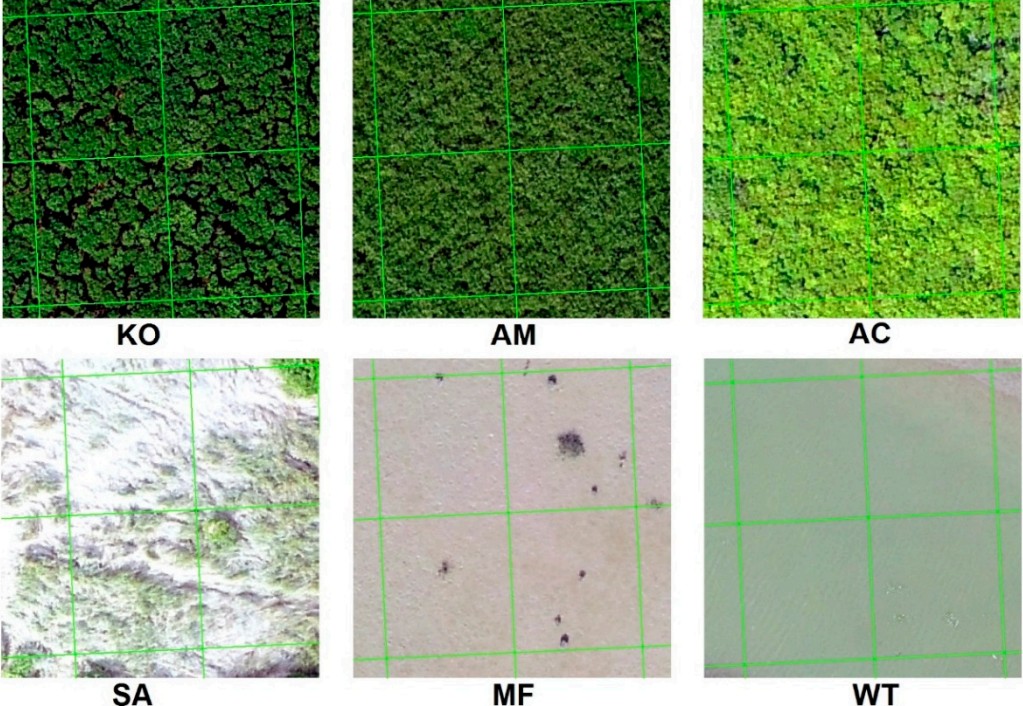

**Figure 4.** Examples of training samples selected by the adjusted fishnet and UAV images. KO: *Kandelia Obovata*, AC: *Aegiceras corniculatum*, AM: *Avicennia marina*, SA: *Spartina alterniflora*, WT: water, and MF: mudflats.

### 2.4.2. Random Forest Classification and Feature Importance

The random forest (RF) classification, as a non-parametric ensemble classification algorithm, has received an increasing amount of interest, because it is more accurate and robust for land cover classifications than tradition classifiers [27,28]. The random forest classifier is composed of a cluster of decision trees, each tree is established by random samples selected independently from the input samples, and the input sample will be classified to the most popular class voted by all trees in the forest [29]. There are several advantages in applying RF algorithm to remote sensing classification researches [27]. First, it is efficient to calculate large databases. Second, as an ensemble algorithm, it is robust to noise and outliers of input data [30]. Moreover, it provides quantitative evaluation of the importance of each input features [27,31].

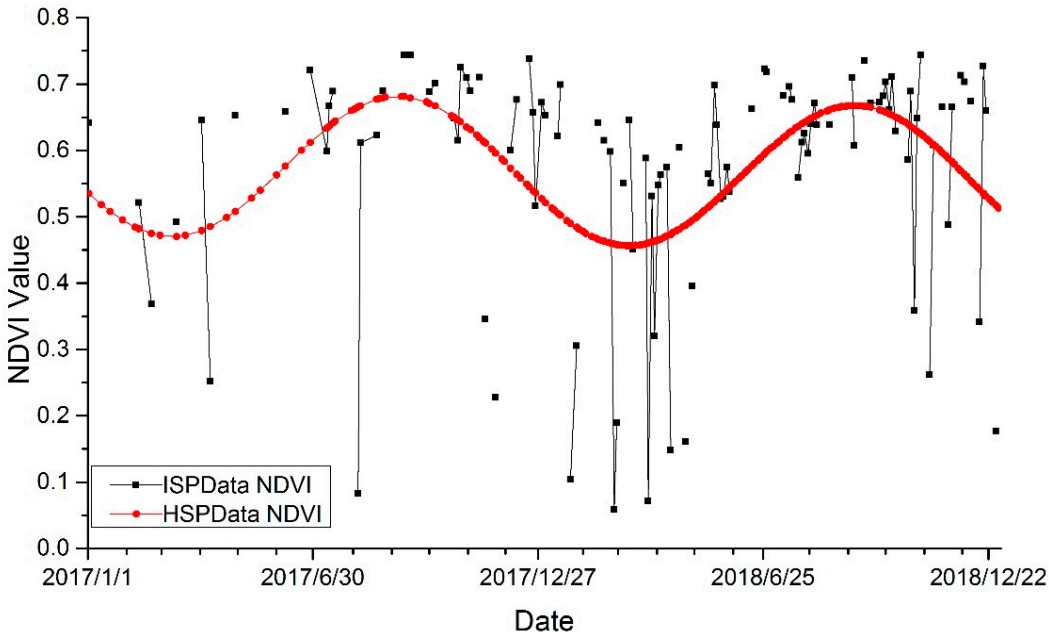

**Figure 5.** Time series Sentinel normalized difference vegetation index (NDVI) profile before (initial Sentinel phenological dataset, ISPData) and after (high-quality Sentinel phenological dataset, HSPData) the HANTS (harmonic analysis of time series) algorithm.

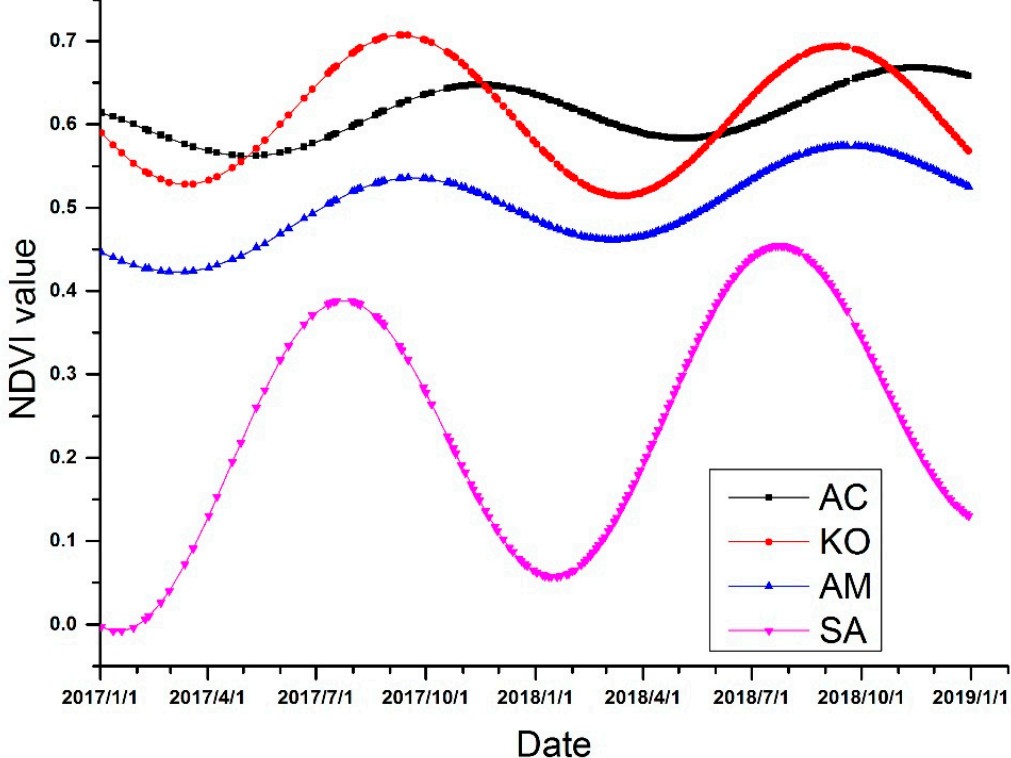

**Figure 6.** Typical time series NDVI profile established by harmonic analysis of time series (HANTS).

The RF algorithm was operated in the GEE platform. The RF algorithm requires two tuning parameters, one is the number of trees (ntree) that in the forest, the other is the number of node in each tree (mtry) [32]. This study set ntree to 500, a number large enough to obtain an unbiased estimate of generalization errors. The number of node, mtry, which largely impacts on the prediction error was set to 5, based on cross-validation [33].

The theory of random forests algorithm in classification of three mangrove species is as follows:

1.  Draw 500 (ntree) bootstrap samples from HSPData.
2.  For each of the bootstrap samples, grow an unpruned classification or regression tree with 5 (mtry) node.
3.  Predict classification result by aggregating the majority votes of the 500 trees.
4.  The RF classifier measures the importance of a feature with respect to the classes by Gini Index. The Gini index can be written as:

$$Gini = \sum \sum_{j \neq i} (f(C_i, T)/|T|)(f(C_j, T)/|T|), \tag{1}$$

where T is a given training set, f ($C_i$, T)/|T| is the likelihood that a selected case (pixel) belongs to class $C_i$. This index is beneficial for studies using multi-source datasets, which contain high dimensional data. It can be used to assure how each input feature influences the classification accuracy, and help to select the features with high importance [27,31]. To evaluate the importance of a certain feature, the RF changes this input features while leaving the rest features constant, and then measures the decrease of the Gini index and mean decrease in accuracy (MDA). The decrease is called the MeanDecreaseGini (MDG), the higher the MDG is, the more important the feature is [28]. MDA is the difference in prediction accuracy before and after permuting variable, averaged over all trees. A high decrease of MDG indicates the importance of that variable [34]. In this study, the MDG and MDA were implemented with the randomForestSRC 2.9.0 package by Ishwaran and Kogalur in 2019. The original classification dataset contained 199 bands for 24 months. To avoid redundant data and noise, 24 cloud free NDVI images were selected in each month to represent 24 months. This dataset was used to validate important months in mangrove species discrimination with the MDG and MDA.

## 3. Results

### 3.1. Classification Map and Accuracy Assessment

The phenology-based mangrove species map is shown in Figure 7. In Zhangjiang estuary, almost all the mangrove forests were located along the west intertidal zone. It is easy to discerned woody canopies from other non-vegetation covers as well as herbaceous SA. As shown in Figure 7, KO and AC were widely distributed in northern part over the region of mangrove forests. AM was mainly situated in the southern part of the mangrove forest. In this study, the accuracy of this map was validated based on ground truth points and a confusion matrix.

Table 1 shows our confusion matrix, which illustrates how good the classification results consistent with the ground truth samples. The map of mangrove species in Zhangjiang estuary was of good quality, as the overall accuracy was 84% and Kappa coefficient was 0.84 (Table 1).

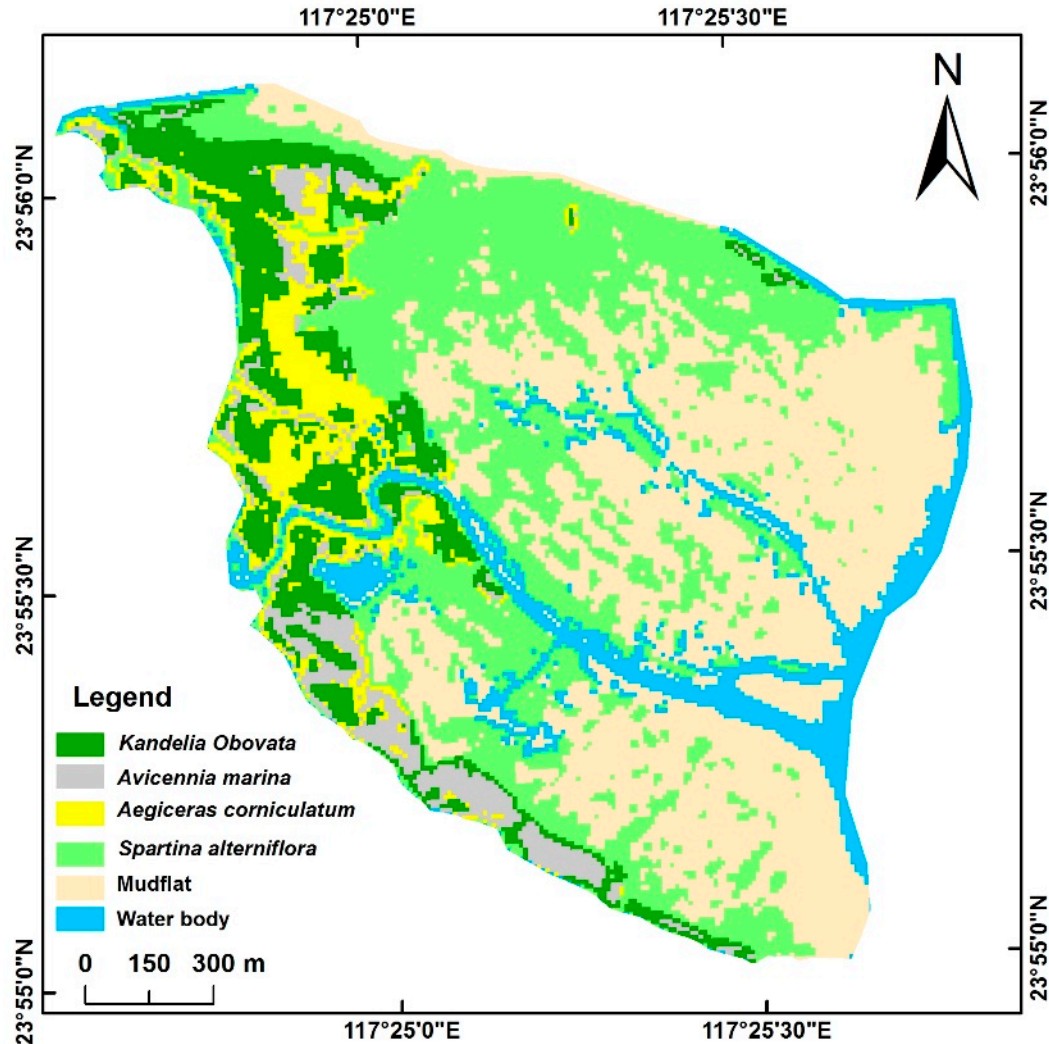

**Figure 7.** Phenology-based mangrove species map including the spatial distribution of different mangrove species and other types of land cover.

**Table 1.** Confusion matrix, overall accuracy, producer's accuracy, user's accuracy, and Kappa coefficient for all classes. KO: *Kandelia Obovata*, AC: *Aegiceras corniculatum*, AM: *Avicennia marina*, SA: *Spartina alterniflora*, WT: water, and MF: mudflats.

| Land Cover | Classification Results | | | | | | |
|---|---|---|---|---|---|---|---|
| | AC | AM | KO | SA | WT | MF | Producer's Accuracy |
| AC | 39 | 2 | 5 | 2 | 0 | 0 | 0.81 |
| AM | 2 | 42 | 0 | 3 | 0 | 1 | 0.88 |
| KO | 7 | 2 | 39 | 4 | 0 | 0 | 0.75 |
| SA | 3 | 1 | 2 | 41 | 2 | 4 | 0.77 |
| WT | 0 | 0 | 0 | 0 | 48 | 2 | 0.96 |
| MF | 1 | 0 | 0 | 3 | 2 | 43 | 0.88 |
| User's accuracy | 0.75 | 0.89 | 0.85 | 0.77 | 0.92 | 0.86 | - |
| Overall accuracy | | 84% | | Kappa coefficient | | | 0.84 |

### 3.2. Important Months in Mangrove Species Detection

Table 2 shows the importance of the contribution of each month to the RF classification of mangrove species. The higherMDGand MDA value indicates that the corresponding month contributed more to the RF classification model. As shown in Table 2, the most influential months included April 2017,

January 2017, March 2018, December 2017, February 2018, February 2017, and January 2018, with MDG values all above 10. All of the months contained valuable information for discriminating among mangrove species. However, according to the cross-validation accuracy (Figure 8), the top nine months raised the accuracy to 83% (Point A), and the top 22 months reached the highest accuracy. The month importance measure indicated that the months in late winter and early spring played critical roles in mangrove species discrimination.

**Table 2.** Variable importance contribution of different months in terms of the MeanDecreaseGini (MDG) index and mean decrease in accuracy (MDA), ranging by importance.

| No. | Month | MDG | MDA | No. | Month | MDG | MDA |
|-----|-------|-----|-----|-----|-------|-----|-----|
| 1 | April 2017 | 13.99 | 0.08 | 13 | August 2017 | 8.82 | 0.03 |
| 2 | January 2017 | 12.48 | 0.08 | 14 | November 2017 | 8.40 | 0.03 |
| 3 | March 2018 | 11.38 | 0.06 | 15 | December 2018 | 8.27 | 0.03 |
| 4 | December 2017 | 11.34 | 0.06 | 16 | October 2017 | 6.36 | 0.02 |
| 5 | February 2018 | 10.81 | 0.05 | 17 | September 2017 | 6.35 | 0.02 |
| 6 | February 2017 | 10.63 | 0.05 | 18 | October 2018 | 6.25 | 0.01 |
| 7 | January 2018 | 10.07 | 0.05 | 19 | August 2018 | 5.53 | 0.01 |
| 8 | November 2018 | 9.85 | 0.04 | 20 | September 2018 | 4.93 | 0.01 |
| 9 | March 2017 | 9.66 | 0.04 | 21 | July 2018 | 4.72 | 0.01 |
| 10 | May 2017 | 9.43 | 0.04 | 22 | June 2018 | 4.25 | 0.00 |
| 11 | July 2017 | 9.01 | 0.03 | 23 | April 2018 | 3.92 | 0.00 |
| 12 | June 2017 | 8.99 | 0.03 | 24 | March 2018 | 3.89 | 0.00 |

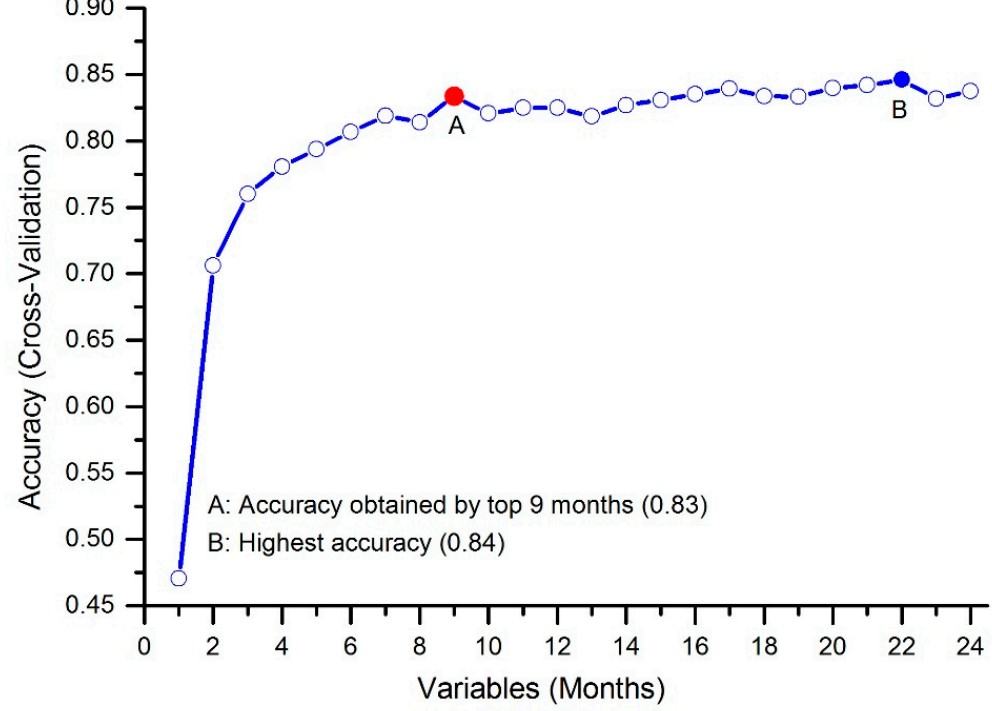

**Figure 8.** Effect of the number of months on the cross-validation accuracy.

To assess the role of the top nine important months, the NDVI bands of these months were classified by RF classification algorithm and training and testing samples described in Section 2.3. Compared to all NDVI bands classification, the overall accuracy of the top nine months decreased by 2% (from 84% to 82%), and the Kappa coefficient decreased by 0.6 (from 0.84 to 0.78; Table 3).

**Table 3.** Confusion matrix of the top nine months classification, including overall accuracy, producer's accuracy, user's accuracy, and Kappa coefficient for all classes. KO: *Kandelia Obovata*, AC: *Aegiceras corniculatum*, AM: *Avicennia marina*, SA: *Spartina alterniflora*, WT: water, and MF: mudflats.

| Land Cover | Classification Results | | | | | | |
|---|---|---|---|---|---|---|---|
| | **AC** | **AM** | **KO** | **SA** | **WT** | **MF** | **Producer's Accuracy** |
| AC | 38 | 3 | 5 | 3 | 0 | 0 | 0.77 |
| AM | 3 | 42 | 0 | 4 | 0 | 1 | 0.84 |
| KO | 7 | 2 | 39 | 4 | 0 | 0 | 0.75 |
| SA | 3 | 2 | 3 | 39 | 2 | 4 | 0.74 |
| WT | 0 | 0 | 0 | 0 | 48 | 2 | 0.96 |
| MF | 1 | 0 | 0 | 3 | 2 | 43 | 0.88 |
| User's accuracy | 0.73 | 0.86 | 0.83 | 0.74 | 0.92 | 0.86 | - |
| Overall accuracy | | 82% | | Kappa coefficient | | | 0.78 |

## 4. Discussion

### 4.1. Comparison with a UAV-Based Classification Map

In this study, our phenology-based mangrove species map was compared with a published UAV-based high resolution map generated by Zhu et al. in 2019 [24]. The UAV-based high resolution map (Figure 9) was shown that for mangrove species mean user's and producer's accuracies were 86.9% and 83.6%, respectively [24]. As shown in Figures 7 and 9, spatial distributions of mangrove species in these two maps were similar. However, in these two maps, the areal proportions of the three mangrove species were different. In the UAV-based map, KO, AM, and AC had real proportions of 67.48%, 12.12%, and 20.4%, respectively, while in the phenology-based map, they were 48.31%, 22.51%, and 29.16%, respectively. Obvious differences could be found in zones (A) and (B). In both of these zones, the phenology-based map had more AM and less AC than the UAV-based map. These differences were mainly caused by the spatial resolution of classification images, as Sentinel only had a spatial resolution of 10 m, while the UAV images had a 5 cm spatial resolution. According to our field survey, in zone A and B, AM and AC were highly mixed together, and the canopy of AM (height 2–3 m, and radius 1–2 m) was higher and wider than AC's (height 0.5–2 m, and radius 0.5–1.5 m). Therefore, in 10 m spatial resolution Sentinel images, trees of AM and AC were mixed in one pixel and mainly showed phenological characters of AMs, while in 5 cm spatial resolution UAV images, most AC and AM could be separated.

To our knowledge, the phenology-based mangrove species map (Figure 7) was the first local mangrove species map that was generated by satellite images. At the same time, the phenological trajectory of local mangrove species (Figure 6) presented by time series NDVI could assist and support local ecologists in mangrove physiology and ecology researches. Moreover, it was also the first attempt to map mangrove species based on plant phenological trajectories obtained from remote sensing images. The methodology of this study offers a great benefit to remote sensing communities in accurately mapping the extent and condition of mangrove forests as well as other highly mixed ecosystems.

### 4.2. Advantages and Limitations of Phenology-Based Classification

S2 provides a high spatial resolution and high temporal frequency of observations and can effectively meet the requirements of phenology-based approaches for patchy mangrove species. In addition, instead of the selection of less cloudy/shadow scenes, we selected individual pixels of good observations to enrich the phenological information. Therefore, each pixel in the study area had 102 to 111 good time series observations (Figure 2) over the 24-month period. The high temporal frequency of observations was sufficient to establish reliable phenological trajectories of different mangrove species. Figure 6 shows the typical NDVI time series profiles of KO, AC, AM, as well as SA. This sentinel based NDVI time series data, compared to the commonly used Landsat based time series data,

presented great improvement. On one hand, Sentinel imagery has a higher spatial resolution than Landsat imagery. On the other hand, Landsat based time series data could, at most, present 24 good observations per year, which may not be sufficient to track the phenological differences in mangrove species [15,16].

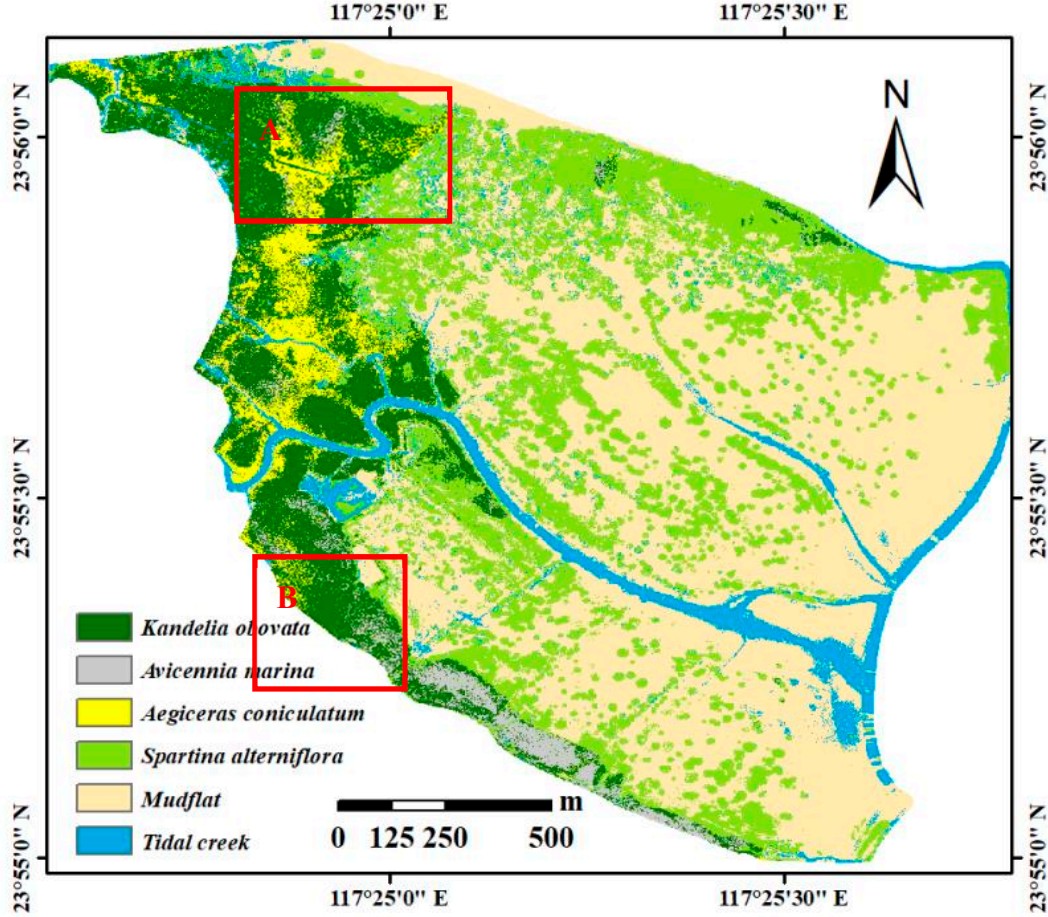

**Figure 9.** UAV-based high resolution mangrove species map [26].

Though our phenology-based approach indicated the feasibility of discriminating mangrove species in a relatively small area of the Zhangjiang estuary, in large scale mangrove species mapping, the influence of within-class phenological variation should be seriously considered. Several reports have shown that mangrove phenology is affected by a combination of abiotic and biotic factors, such as the regional climate, seawater and soil salinity, latitude, and local vegetation interactions in ecological systems [35–37]. Thus, the same mangrove species may appear to have considerable different phenological characteristics in different locations [35–37]. Therefore, it is difficult to determine the phenological trajectory of a mangrove species across different locations and years without expert knowledge. Thus, the limitation of our mangrove species detection strategy is the uncertainties in carrying out long-term, large-scale monitoring.

### 4.3. Advantages of Using Training Samples from UAV Images

The random forest classification, as one of the supervised classification algorithms, is greatly affected by the quality of the labeled samples used to train the classifier. The reliability of training samples is of fundamental importance to allow adequate learning of the properties of the investigated image and then to design the rules in a classifier [38]. Thus, correct training samples are essential for obtaining highly accurate classification maps. However, in real cases, the correction of the training

samples cannot be assured for two reasons. First, the bias of the positioning system leads to training samples with incorrect locations and thus, wrong pixels in the remotely sensed image. Second, the transformation of training samples (e.g., field survey points) to pixels with only one land cover type (pure pixel) is difficult. For example, in this study, a training sample should at least represent a 10 m × 10 m pure pixel. Traditionally, to assure that a 10 m × 10 m pure pixel is obtained, a 20 m × 20 m pure plot should be collected during ground surveys. However, due to the small area and highly mixed mangrove species in Zhangjiang estuary, there is a limited number of 20 m × 20 m pure plots. Meanwhile, it is difficult to arrange field plots due to the muddy environment of mangrove swamps and the high density of mangrove canopies.

To solve the aforementioned issues, a vital feature of the present study was the use of the adjusted fishnet and UAV images for the collection of quality training samples. Each plot in the adjusted fishnet was set to fit a pixel in the S2 image (10 m × 10 m). Then, all plots were applied to the high-resolution UAV image. These solutions provided the maximum number of high-quality ground truth samples. Additionally, these solutions are cost-effective and labor-saving.

### 4.4. Benefits from Using the Google Earth Engine (GEE) Platform

In this study, GEE-based cloud computing effectively facilitated the processing and classification of hundreds of Sentinel data points for mangrove species mapping. During image processing procedures, GEE synchronized all the S2 data from the European Space Agency and easily removed cloudy pixels and then established high-quality time-series NDVI data using simple codes. Moreover, as many machine-learning algorithms were already integrated into the GEE Application Programming Interface (API), it was convenient to apply the random forest classification on the GEE platform. Thus, all procedures in our phenology-based classification could be operated on the cloud computing platform, not subject to local computer or other software. Information extraction from remote sensing data was therefore simplified.

## 5. Conclusions

This study indicated the feasibility and reliability of mapping mangrove species in Zhangjiang estuary using an NDVI time series, which was built based on S2 imagery. The GEE was used to build and reconstruct dense time series data and process the random forest classification. To our knowledge, this study was the first investigation to map mangrove species based on plant phenological trajectories. The implementation of this study was attributed to several factors including the improved data (S2 with 10 m spatial resolution and a revisit interval of 2–5 days), platform (GEE), and algorithm (RF) and the consistent natural conditions (ground surveys and UAV images). There were several findings, as follows:

1. In Zhangjiang estuary, there were mainly three types of mangrove species (AC, AM, and KO), and one type of invasive species (SA). To build the NDVI time series, we collected 199 scenes of S2 images from 1 January 2017 to 31 December 2018. After removing noise and filling gaps in the initial NDVI time series by the HANTS algorithm, we found that the phenological trajectories of AC, AM, KO, and SA showed great differences (Figure 6).
2. The random forest algorithm was applied to the NDVI time series, and the overall accuracy and Kappa confidence of the mangrove species map were 84% and 0.84, respectively. To acquire sufficient high-quality training and validation samples, UAV images were adopted to give pure pixels of different mangrove species.
3. The feature importance measurement showed that the months in late winter and early spring played critical roles in mangrove species discrimination. Phenological signatures of nine months (April 2017, January 2017, March 2018, December 2017, February 2018, February 2017, January 2018, November 2018, and March 2017) increased the overall accuracy to 83%.

The findings of this study made a valuable contribution to the classification of mangrove species using medium resolution satellite images. However, because the phenological trajectory of a given mangrove species may vary in different locations, future work should address the relationship between the phenological trajectory and the local environment conditions.

**Author Contributions:** H.L. and M.J. designed the study, processed the data, and wrote the manuscript draft. R.Z., Y.R. and X.W. helped with the image analysis and fieldwork.

**Funding:** The work was supported by the National Natural Science Foundation of China (No. 41601470, No. 41601406), the Strategic Planning Project of the Institute of Northeast Geography and Agroecology (IGA), Chinese Academy of Sciences (No. Y6H2091000), and the Youth Innovation Promotion Association of Chinese Academy of Sciences (2017277, 2012178). This work was supported by the Open Fund of State Laboratory of Information Engineering in Surveying, Mapping and Remote Sensing, Wuhan University (Grant No. 19I02). Northeast Branch of National Earth System Science Data Center (http://northeast.geodata.cn). The Science and Technology Basic Resources Investigation Program of China (No. 2017FY100706).

**Acknowledgments:** The authors are grateful to the colleagues who participated in the field surveys and data collection.

**Conflicts of Interest:** The authors declare no conflict of interest.

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
