# Peer review of "Incorporating the Plant Phenological Trajectory into Mangrove Species Mapping with Dense Time Series Sentinel-2 Imagery and the Google Earth Engine Platform"

_remotesensing, doi:10.3390/rs11212479_

Round 1

Reviewer 1 Report

Paper covers an interesting issue of vegetation phenological trajectories, however these trajectories are in my opinion insufficiently explored and described. 

Major comments:

As the “phenoloical trajectories” is a part of this study and the term is even part of the title, I think that it should be more extensively studied in this paper. For example, more focus should be put on phenological trajectories of each species, e.g. how they change during the year an why and which time of the year is crucial for these species discrimination?

Maybe variable importance should be performed also with other methods (e.g. like Mean Decrease Accuracy, Variable Importance using Random Forest). I think it will be also interesting to test which months are important for classification of particular species.

I would suggest to test the classification of the reduced data sets also, i.e. imagery from months with the highest contribution, like top 9 months.

Minor comments:

In my opinion table 1 with the characteristics of S2 MSI is unnecessary - Sentinel-2 it's not brand new mission and it's currently widely used.

In line 287 do you mean figure 6?

Author Response

Reviewer 1:

As the “phenoloical trajectories” is a part of this study and the term is even part of the title, I think that it should be more extensively studied in this paper. For example, more focus should be put on phenological trajectories of each species, e.g. how they change during the year and why and which time of the year is crucial for these species discrimination?

Response: Thanks for this comment. The phenoloical trajectories” in this study means land surface phenology (LSP). The LSP is the reflectance properties of vegetated land varies seasonally in relation to vegetation phenology. The systematic, multi-temporal data collected by optical satellite sensors offer a unique mechanism to monitor LSP as this approach allows monitoring of an entire ecosystem rather than individual plant.

At the time we preparing this manuscript, we were thinking to add some contents about the phenology of these mangrove species in the “Study area” section. However, we failed to find any literatures or reports about phenology of local mangrove species such as Kandelia Obovata, Aegiceras corniculatum, and Avicennia marina. Moreover, according to international literatures, as pioneer species, the same mangrove species may appear to have considerable different phenological characteristics in different locations.

According to our experiment, different mangrove species in Zhangjiang Estuary do have different phenoloical trajectories. The difference of NDVI time series in various species was significant due to the capacity of cold resistance and tolerance of lack water during the grow seasons, which was benefit to identify mangroves species. Under the influence of higher temperature and heavier rain in summer and autumn, the mangroves grow well characterized by continuous raised NDVI profile. In this study, the three main mangrove species have different phenoloical trajectories. KO shows obvious seasonal trends, because it has high shedding rate. AM and AC have similar shedding rate, but the NDVI values of AC are always higher than AM’s. The scope of this study is to distinguish mangrove species based on the difference of phenoloical trajectories in various species. 

Previous studies of mangrove forests have indicated that litterfall dynamics and reproductive phenology of mangroves is driven by a complex interaction of ecological, climatic, local environmental factors and natural disturbances. Too much research on mangrove phenology was out of the scope of this study. Still thank you so much. In future, with the availability of the data more concern with biological and phenological characters of mangrove species, analysis will be conducted.

Maybe variable importance should be performed also with other methods (e.g. like Mean Decrease Accuracy, Variable Importance using Random Forest). I think it will be also interesting to test which months are important for classification of particular species.

Response: Thanks so much for your comment. Mean Decrease Accuracy was added in both section Methods and Results in this revised manuscript.

In Methods section

Mean Decrease in Accuracy (MDA) criteria was used for measuring the variables importance of explanatory power of the phenoloical trajectories bands in the classification of three mangrove species. The MDA criterion was rendered as an output of RF in R software.

To evaluate the importance of a certain feature, the RF changes this input features while leaving the rest features constant, and then measures the decrease of the Gini Index and Mean Decrease in Accuracy (MDA). The decrease is called MeanDecreaseGini (MDG), the higher the MDG is, the more important the feature is [26]. MDA is the difference in prediction accuracy before and after permuting variable, averaged over all trees. A high decrease of MDG indicates the importance of that variable [32]. In this study, the MDG and MDA were implemented with the random Forest SRC 2.9.0 package by Ishwaran and Kogalur in 2019.

   In results section

Table 2. Variable importance contribution of different months in terms of the MeanDecreaseGini (MDG) Index and Mean Decrease in Accuracy (MDA), ranging by importance.

No.

Month

MDG

MDA

No.

Month

MDG

MDA

1

Apr. 2017

13.99

0.08

13

Aug. 2017

8.82

0.03

2

Jan. 2017

12.48

0.08

14

Nov. 2017

8.40

0.03

3

Mar. 2018

11.38

0.06

15

Dec. 2018

8.27

0.03

4

Dec. 2017

11.34

0.06

16

Oct. 2017

6.36

0.02

5

Feb. 2018

10.81

0.05

17

Sep. 2017

6.35

0.02

6

Feb. 2017

10.63

0.05

18

Oct. 2018

6.25

0.01

7

Jan. 2018

10.07

0.05

19

Aug. 2018

5.53

0.01

8

Nov. 2018

9.85

0.04

20

Sep. 2018

4.93

0.01

9

Mar. 2017

9.66

0.04

21

Jul. 2018

4.72

0.01

10

May 2017

9.43

0.04

22

Jun. 2018

4.25

0.00

11

Jul. 2017

9.01

0.03

23

Apr. 2018

3.92

0.00

12

Jun. 2017

8.99

0.03

24

Mar. 2018

3.89

0.00

I would suggest to test the classification of the reduced data sets also, i.e. imagery from months with the highest contribution, like top 9 months.

Response: Thanks for this comment, it is a great idea. We added the following contents as you suggested in the manuscript.

To assess the role of the top 9 important months, the NDVI bands of these months’ were classified by RF classification algorithm and training and testing samples described in section 2.3. Compared to all NDVI bands classification, the overall accuracy of top 9 months’ decreased by 2% (from 84% to 82%), and the Kappa coefficient decreased by 0.6 (from 0.84 to 0.78) (Table3).

Table 3. Confusion matrix of top 9 months classification, including overall accuracy, producer’s accuracy, user’s accuracy, and Kappa coefficient for all classes. KO: Kandelia Obovata, AC: Aegiceras corniculatum, AM: Avicennia marina, SA: Spartina alterniflora, WT: water, MF: mudflats.

Land

cover

Classification results

AC

AM

KO

SA

WT

MF

Producer’s accuracy

AC

38

3

5

3

0

0

0.77

AM

3

42

0

4

0

1

0.84

KO

7

2

39

4

0

0

0.75

SA

3

2

3

39

2

4

0.74

WT

0

0

0

0

48

2

0.96

MF

1

0

0

3

2

43

0.88

User’s accuracy

0.73

0.86

0.83

0.74

0.92

0.86

--

Overall accuracy

82%

Kappa  coefficient

0.78

Minor comments:

In my opinion table 1 with the characteristics of S2 MSI is unnecessary - Sentinel-2 it's not brand new mission and it's currently widely used.

Response: Thanks for this comment, we agreed with you, the table was deleted.

In line 287 do you mean figure 6?

Response: Thanks for this comment. It has been modified to Figure 6.

Reviewer 2 Report

This study used Sentinal-2 imagery to build up fine resolution NDVI time series to classify coastal mangrove species for mapping based on a Random Forest algorithm. Ground-based survey and UAV imagery were also used as reference data to validate the classification. The results were generally solid, though many details of the approach mechanisms and how different components connected in the approach were not clearly explained. The discussion should also be improved to explain the contributions of this study.

One major issue is although the whole manuscript mentioned many times about phenology-based approach, there was no clear or direct explanation on the approach. How did NDVI time series data carry phenological information of different species that can be classified by the RF algorithm? What were exact phenological information used in the classification? If the RF algorithm classified the time series based on the timing of lowest or highest peak points or the inflection points or the decreasing or increasing patterns in the time series, how did these information reflect the differences among mangrove species phenologically and biologically? The mechanism of the phenology-based approach should be much more clearly explained and discussed.   

Other issues:

Why used NDVI? Not EVI or other vegetation index?

Was RF classification based on two-year NDVI time series of all pixels? Should clearly explain how RF algorithm worked on NDVI time series data. Should move line 245-248 to Method section.

Figure 9. How was the UAV-based classification map generated? Using the same RF algorithm or different method? Should clearly explain this in Methods and Results.

Discussion section should explain and highlight the contribution of the research findings in this study to the field or other research. As a methodology study, should better focus on the application into other relevant research and implication for further development.

Line 273-275. Should clearly explain why different spatial resolution lead to different classification results in A and B areas. Why most difference happened in A and B areas?

Line 298-302. How do you overcome the limitation in application of the approach in a long-term and large-scale situation?

Was there any field-based phenology observation or empirical prior knowledge on mangrove phenology of three species to validate the classification?

Author Response

Reviewer 2:

This study used Sentinal-2 imagery to build up fine resolution NDVI time series to classify coastal mangrove species for mapping based on a Random Forest algorithm. Ground-based survey and UAV imagery were also used as reference data to validate the classification. The results were generally solid, though many details of the approach mechanisms and how different components connected in the approach were not clearly explained. The discussion should also be improved to explain the contributions of this study.

One major issue is although the whole manuscript mentioned many times about phenology-based approach, there was no clear or direct explanation on the approach. How did NDVI time series data carry phenological information of different species that can be classified by the RF algorithm? What were exact phenological information used in the classification? If the RF algorithm classified the time series based on the timing of lowest or highest peak points or the inflection points or the decreasing or increasing patterns in the time series, how did these information reflect the differences among mangrove species phenologically and biologically? The mechanism of the phenology-based approach should be much more clearly explained and discussed.

Response: Thanks for this comment. The volatilities in time series Sentinel NDVI profile of mangroves could be mainly explained by litterfall dynamics. From January to February, minimum values of litterfall were found during the cold and dry season, while the peak in litterfall corresponded to the end of dry season and beginning of rainy season in April. Mangrove trees shed leaves, potentially with a decrease in leaf cover resulting in a trough in NDVI in April. The difference of NDVI time series in various species was significant due to the capacity of cold resistance and tolerance of lack water during the grow seasons, which was benefit to identify mangroves species. Under the influence of higher temperature and heavier rain in summer and autumn, the mangroves grow well characterized by continuous raised NDVI profile. In this study, the three main mangrove species have different phenoloical trajectories. KO shows obvious seasonal trends, because it has high shedding rate. AM and AC have similar shedding rate, but the NDVI values of AC are always higher than AM’s.

     Some sentences have been added to describe the phenological information of NDVI time series, please see line 188 to 194.      

Studies of mangrove forests have indicated that litterfall dynamics and reproductive phenology of mangroves is driven by a complex interaction of ecological, climatic, local environmental factors and natural disturbances. Too much research on mangrove phenology was out of the scope of this study. In future, with the availability of the data more concern with biological and phenological characters of mangroves, analysis will be conducted.

The random forest (RF) algorithm is a non-parametric ensemble classification algorithm. RF classifier is composed of a cluster of decision trees, each tree is established by random samples selected independently from the input samples (in this study is time series NDVI), and the input sample will be classified to the most popular class voted by all trees in the forest. So the RF algorithm classify mangrove species were not based on certain lowest or highest peak points or the inflection points, it treated every point equally. However, RF could give the variable importance after classification. More information about how RF algorithm classify mangrove species based on NDVI data can be found in the response of the third comment.

Other issues:

Why used NDVI? Not EVI or other vegetation index?

Response: Thanks for this comment. In this study we chose NDVI, because it is most widely used vegetation index, and time series changes in NDVI have long been used to represent vegetation phenology (Diao et al., 2016; Reed et al., 1994). NDVI allows comparison with previous studies (Pastor-Guzman et al., 2018). Moreover, according to our experiments, time series NDVI could satisfy the objectives of this study.

EVI is also a great choice, but atmospheric corrected images (Bottom Of Atmosphere images) were strictly needed when calculating EVI. However, according to official website Bottom Of Atmosphere images (Level 2A) of Sentinel 2 contained certain error and were suggested not to use. Other vegetation index were rarely used in mangrove phenology research.

References:

Diao, C.; Wang, L. Incorporating plant phenological trajectory in exotic saltcedar detection with monthly time series of Landsat imagery. Remote Sensing of Environment 2016, 182, 60-71.

Reed, B.C.; Brown, J.F.; VanderZee, D.; Loveland, T.R.; Merchant, J.W.; Ohlen, D.O. Measuring phenological variability from satellite imagery. Journal of vegetation science 1994, 5, 703-714.

Pastor-Guzman, J.; Dash, J.; Atkinson, P.M. Remote sensing of mangrove forest phenology and its environmental drivers. Remote Sensing of Environment 2018, 205, 71-84, doi:10.1016/j.rse.2017.11.009.

Was RF classification based on two-year NDVI time series of all pixels? Should clearly explain how RF algorithm worked on NDVI time series data. Should move line 245-248 to Method section.

Response: Thanks for this comment. The RF classification was based on high-quality Sentinel phenological dataset. This dataset contains two-year NDVI time series of all pixels (199 bands for 24 months) smoothed by the Harmonic ANalysis of Time Series (HANTS) algorithm. The RF classification was carried out in GEE planform.

The reduced dataset (24 cloud free NDVI images represent 24 months) were only used to validate the importance of variable (calculating Mean Decrease in Accuracy and MeanDecreaseGini). This was implemented with the randomForestSRC 2.9.0 package in R software. Because it is hard to calculate Mean Decrease in Accuracy and MeanDecreaseGini in GEE planform.

Line 245-248 were moved to Method section, and we also added contents about how RF algorithm worked on NDVI time series data, as follows:

The theory of random forests algorithm in classification of three mangrove species is as follows:

Draw 500 (ntree) bootstrap samples from HSPData. For each of the bootstrap samples, grow an unpruned classification or regression tree with 5 (mtry) node. Predict classification result by aggregating the majority votes of the 500 trees.

To evaluate the importance of a certain feature, the RF changes this input features while leaving the rest features constant, and then measures the decrease of the Gini Index and Mean Decrease in Accuracy (MDA). The decrease is called MeanDecreaseGini (MDG), the higher the MDG is, the more important the feature is [26]. MDA is the difference in prediction accuracy before and after permuting variable, averaged over all trees. A high decrease of MDG indicates the importance of that variable [32]. In this study, the MDG and MDA were implemented with the randomForestSRC 2.9.0 package by Ishwaran and Kogalur in 2019. The original classification dataset contained 199 bands for 24 months. To avoid redundant data and noise, 24 cloud free NDVI images were selected in each month to represent 24 months. This dataset was used to validate important months in mangrove species discrimination with the MDG and MDA.

Figure 9. How was the UAV-based classification map generated? Using the same RF algorithm or different method? Should clearly explain this in Methods and Results.

Response: Thanks for this comment. The UAV-based classification map was not generated by this study, it is a published result by Zhu et al., 2019. To explain this we added a new sentence in the revised manuscript.

In this study, our phenology-based mangrove species map was compared with a published UAV-based high resolution map generated by Zhu et al. in 2019 [24].

References:

Zhu, X.; Hou, Y.; Weng, Q.; Chen, L. Integrating UAV optical imagery and LiDAR data for assessing the spatial relationship between mangrove and inundation across a subtropical estuarine wetland. ISPRS Journal of Photogrammetry and Remote Sensing 2019, 149, 146-156, doi:10.1016/j.isprsjprs.2019.01.021.

Discussion section should explain and highlight the contribution of the research findings in this study to the field or other research. As a methodology study, should better focus on the application into other relevant research and implication for further development.

Response: Thanks for this comment. Discussion section has been strengthened as follows:

To our knowledge, the phenology-based mangrove species map (Figure 7) is the first local mangrove species map that generated by satellite images. At the same time, the phenological trajectory of local mangrove species (Figure 6) presented by time series NDVI can assist and support local ecologists in mangrove physiology and ecology researches. Moreover, it is also the first attempt to map mangrove species based on plant phenological trajectories obtained from remote sensing images. The methodology of this study offers great benefit remote sensing communities in accurately mapping the extent and condition of mangrove forests as well as other highly mixed ecosystems using medium resolution satellite images.

However, because the phenological trajectories of a given mangrove species may vary in different locations, future work should address the relationship between the phenological trajectory and the local environment conditions.

Line 273-275. Should clearly explain why different spatial resolution lead to different classification results in A and B areas. Why most difference happened in A and B areas?

Response: Thanks for this comment. We explained this in the revised manuscript, as following.

Line 301-306

According to our field survey, in zone A and B, AM and AC are highly mixed together, and the canopy of AM (height 2-3 m, radius 1-2 m) is higher and wider than AC’s (height 0.5-2 m, radius 0.5-1.5 m). Therefore, in 10 m spatial resolution Sentinel images, trees of AM and AC were mixed in one pixel and mainly showed phenological characters of AM’s, while in 5 cm spatial resolution UAV images, individual trees of AC and AM can be separated.

Line 298-302. How do you overcome the limitation in application of the approach in a long-term and large-scale situation?

Response: Thanks for this comment. The limitation in application of the approach is that same mangrove species may appear to have considerable different phenological characteristics in different locations. So to overcome the limitation in long-term and large-scale application, export knowledge is needed when training RF algorithm. Moreover, further researches and investigations are also needed to explain changes of phenological variation with local environments, so that, we could predict phenological trajectories of mangrove species by knowing abiotic and biotic factors. 

Was there any field-based phenology observation or empirical prior knowledge on mangrove phenology of three species to validate the classification?

Response: Thanks for this comment. Unfortunately, we didn’t have any field-based phenology observations. However, we do have empirical prior knowledge based on many times of field surveys. The three main mangrove species have distinct color characteristics in spring and early summer (dark-green for KO, grey for AM, and yellow-green for AC). In July to October, all these mangroves have higher Leaf Area Index than other seasons. These characteristics are consisted with NDVI derived phenological trajectories. In future, we will carry out some field-based phenology surveys to study into the biological and phenological characters of mangrove species. Thank you so much.  

Round 2

Reviewer 1 Report

Presented study has been improved in comparison to the first version. It presents interesting and important topic.